# Impact of Pyrolysis Temperature on the Physical and Chemical Properties of Non-Modified Biochar Produced from Banana Leaves: A Case Study on Ammonium Ion Adsorption

**DOI:** 10.3390/ma17133180

**Published:** 2024-06-28

**Authors:** Fernanda Pantoja, Sándor Beszédes, Tamás Gyulavári, Erzsébet Illés, Gábor Kozma, Zsuzsanna László

**Affiliations:** 1Doctoral School of Environmental Sciences, University of Szeged, H-6720 Szeged, Hungary; fernanda.pantoja@mk.u-szeged.hu; 2Department of Process Engineering, University of Szeged, H-6725 Szeged, Hungary; beszedes@mk.u-szeged.hu; 3Department of Applied and Environmental Chemistry, Institute of Chemistry, University of Szeged, H-6720 Szeged, Hungary; gyulavarit@chem.u-szeged.hu (T.G.); kozmag@chem.u-szeged.hu (G.K.); 4Department of Food Engineering, University of Szeged, H-6725 Szeged, Hungary; nyergesne.illes.erzsebet@szte.hu

**Keywords:** biochar, pyrolysis temperature, adsorption, physico-chemical properties

## Abstract

Given the current importance of using biochar for water treatment, it is important to study the physical–chemical properties to predict the behavior of the biochar adsorbent in contact with adsorbates. In the present research, the physical and chemical characteristics of three types of biochar derived from banana leaves were investigated, which is a poorly studied raw material and is considered an agricultural waste in some Latin American, Asian, and African countries. The characterization of non-modified biochar samples pyrolyzed at 300, 400, and 500 °C was carried out through pH, scanning electron microscopy, energy dispersive X-ray spectroscopy, Fourier transform infrared spectroscopy, and specific surface area measurements. The adsorption properties of banana leaf-derived biochar were evaluated by ammonium ion adsorption experiments. The results demonstrated that the pyrolysis temperature has a large impact on the yield, structure, elemental composition, and surface chemistry of the biochar. Biochar prepared at 300 °C is the most efficient for NH_4_^+^ adsorption, achieving a capacity of 7.0 mg of adsorbed NH_4_^+^ on each gram of biochar used, while biochar samples prepared at 400 and 500 °C show lower values of 6.1 and 5.6 mg/g, respectively. The Harkins–Jura isotherm model fits the experimental data best for all biochar samples, demonstrating that multilayer adsorption occurs on our biochar.

## 1. Introduction

Biochar is a carbonaceous material that has gained interest from the scientific community due to its excellent properties and multipurpose benefits for the environment. Most importantly, it can be used to improve the soil and adsorb contaminants in water treatment [1]. Pyrolyzing biomass at high temperatures without oxygen produces gases, bio-oils, and solid biochar. Typical temperatures and residence times used in pyrolysis are in the ranges of 300–900 °C and minutes (fast pyrolysis) to hours (slow pyrolysis), respectively [2]. However, maximum yield is achieved by slow pyrolysis, making it the most widely used method to produce biochar [3].

Different productions and sources result in biochar with different properties and characteristics [4]. In general, the three main objectives for characterizing biochar are the following: (1) to better understand its physical and chemical characteristics, as well as the changes in its characteristics resulting from the different parameters used during production, such as pyrolysis temperature and feedstocks; (2) to evaluate its relevance in the fields of interest; and (3) to study biochar contaminants and ecotoxicological parameters [3]. An extensive variety of biomass types have been considered as raw materials for producing biochar, including agricultural wastes, forestry residues, and animal manure [5].

The use of agricultural waste as biomass is increasingly studied and considered as an alternative environmentally friendly solution to the problems related to its final disposal [6]. One of the crops that generates a large amount of agricultural waste (biomass) is banana, especially in Latin American, Asian, and African countries located in tropical and subtropical zones. Since banana is consumed worldwide throughout the year, its export was estimated to reach 19.2 million tons in 2023 [7]. Additionally, the fruit consumed represents just 21% of the total biomass of the plant; the remaining 79% is biomass treated as waste [8].

Due to its low cost and straightforwardness, adsorption is regarded as one of the most efficient physical ways to eliminate pollutants from water [9]. Its large specific surface area, porous structure, and some specific surface functional groups and mineral components make biochar an effective adsorbent to remove contaminants causing eutrophication from aqueous solutions [4], such as ammonium ions [10].

The physicochemical properties of biochar are significantly influenced by the pyrolysis temperature; previous research has demonstrated that different pyrolysis temperatures result in biochar with different characteristics [11]. The pyrolysis temperature range used in the present research work was determined based on the following considerations: at temperatures above 600 °C, biochar yield may decrease due to the decomposition of volatile fractions and the formation of an intermediate melt in the biochar structure [12]; biochar becomes condensed due to the aromatization of carbon, leading to the crystallization of silicon [13]; the porosity, ash content, electrical conductivity (EC), and pH value of biochar increase with temperature, while cation exchange capacity (CEC), H, C, and N contents decrease [14]; and the surface structure of biochar becomes more complex [15,16,17].

The effects of pyrolysis temperature on the characteristics of banana leaf-derived biochar have not received much attention so far. Accordingly, the objective of the present study is to compare the different characteristics of biochar produced from a poorly studied raw material at three different pyrolysis temperatures. Considering the factor of biochar yield per gram of biomass and the fact that the biochars produced were applied for ammonium adsorption, the temperature range of 300 °C, 400 °C, and 500 °C was chosen. Additionally, the efficiency of each biochar for adsorbing ammonium ions is also investigated.

## 2. Materials and Methods

### 2.1. Lignin, Cellulose, and Ash Content of Banana Leaves

Lignin and cellulose contents in the biomass were determined according to the Chesson method [18]. The calculations were carried out based on Equations (1)–(3) and the meaning of the letters in the equations is explained in the following description. A mixture containing 1 g of ground (particle size 250 μm) and dried banana leaves (a) and 150 mL of distilled water was heated in a water bath at a temperature of 90–100 °C for 1 h. The mixture was filtered through filter paper (size 205 mm, medium filtration rate, particle retention 5–13 μm; VWR International Kft, Debrecen, Hungary), and the residue was washed with 300 mL of hot distilled water. The residue was dried in a vacuum dryer until reaching a constant weight (b). The residue was then mixed with 150 mL of 0.5 M sulfuric acid (H_2_SO_4_) and heated in the water bath at a temperature of 90–100 °C for 1 h. The mixture was filtered and washed using 300 mL of distilled water, followed by drying the residue (c). The dried residue was soaked in 10 mL of 72% H_2_SO_4_ at room temperature for 4 h. Afterwards, 150 mL of 0.5 M H_2_SO_4_ was added to the mixture, which was refluxed in the water bath for 1 h. The solid was washed using 400 mL of distilled water, dried in a vacuum dryer at 105 °C for 30 min, and weighed (d). The solid was heated until it became ash, which was then weighed (e). Finally, the percentages of cellulose, lignin, and ash (*%cellulose*, *%lignin* and *%ash* respectively) were calculated as follows:(1)$$\%\;cellulose=\frac{c-d}{a}*100$$
(2)$$\%\;lignin=\frac{d-e}{a}*100\;$$
(3)$$\%\;ash=\frac{e}{a}*100$$

### 2.2. Biochars Preparation

The banana leaves (*Musa nana*) were provided by the Botanical Garden in Szeged city. The leaves were washed with distilled water several times to remove dust and cut into small pieces. The material was dried in a vacuum dryer at 105 °C for 2 h (Kambic, maximum operating temperature of 200 °C, EU). The dried material was ground and sifted. The material was subjected to pyrolysis at 300, 400, and 500 °C (BC_300, BC_400, and BC_500, respectively) during 2 h in a muffle furnace (Nabertherm, LE 2/11/R6, Germany) under an oxygen-depleted atmosphere situation that was achieved through placing the material in ceramic crucibles covered with aluminium foil and encapsulated in metal crucibles with lids with high temperature resistant. 

Finally, the biochar product was ground in a ceramic mortar and sifted through a mesh test sieve (Endecotts laboratory (England)). The particle size used for the experiments was 250 μm, and the resulting biochar powders were used to determine their physical–chemical characteristics.

### 2.3. Biochar Yield Calculation

The biochar yield was calculated (Equation (4)) considering the weight of biomass entering the muffle furnace and the weight of biochar obtained after pyrolysis:(4)$$\%Yield=\frac{w_{biochar}}{w_{biomass}}*100\;$$
where *w_biochar_* is the weight of biochar (g) obtained after pyrolysis and *w_biomass_* is the weight of raw material (g).

### 2.4. pH Analysis

The pH values of the three different biochars were measured within a solution consisting of 200 mL of distilled water and 500 mg of biochar with an Orion 5 Star pH meter (Thermo Scientific; 9v, Shanghai, China). The solutions were stirred for 30 min at 250 rpm. The pH measurements were also carried out for all adsorption analyses. Therefore, the pH data were recorded once in the initial solution and then once the biochar sample came into contact with the solution following homogenization. In most cases, 500 mg of biochar was used, but other doses were also used when the effect of adsorbent dose was investigated.

### 2.5. Zeta-Potential Analysis

To investigate the surface chemistry of the biochars pyrolyzed at different temperatures, we measured their zeta potentials. For this purpose, 10 mg of each biochar was put in glass test tubes containing 10 mL of 0.01 M sodium chloride (NaCl) solutions with different pH values. After stirring the suspensions, the pH of the samples was measured and adjusted to the different study values (from 3 to 10). Finally, the zeta potential values were measured using electrophoretic light scattering (ELS) (Zetasizer NanoZs instrument, Malvern Panalytical Ltd., Malvern, UK).

### 2.6. Scanning Electron Microscopy and Energy-Dispersive X-ray (SEM–EDX) Analysis

The morphology and elemental composition of each biochar were examined using the integrated secondary electron detector or the Röntec QX2 EDS detector, respectively, of a Hitachi S-4700 Type II microscope (Hitachi, Tokyo, Japan), operated at 10 or 20 kV accelerating voltages.

### 2.7. Fourier Transform Infrared Spectroscopy (FT-IR) Analysis

FT-IR studies were performed with a Bruker Vertex 70 IR spectrometer (Bruker, Billerica, MA, USA; 16 scans/s, 4 cm^−1^ resolution), using the KBr pellet technique. 

### 2.8. Brunauer–Emmett–Teller (BET) Analysis

Each biochar were characterized by adsorption–desorption analysis using N_2_ vapor after preparation of biochars under vacuum for 2 h, at 200 °C to obtain the specific surface area (SSA), total pore volume, and average pore size by a Quantachrome Nova 3000e instrument (Quantachrom Instruments, Munich, Germany).

### 2.9. Sorption Experiments

Stock solutions of ammonium ions were prepared to study the adsorption properties of the biochars. For this purpose, 1000 mg/L [NH_4_^+^] ion concentration stock solution was prepared by dissolving in 1 L of ultrapure water. Ultrapure water with an electrical conductivity of 0.055 µS/cm was obtained through Adrona Crystal equipment (ADRONA SIA, model EX-1101, Latvia) and used for the adsorption experiments.

Preliminary experiments were carried out to determine the biochar dosage, pH of the solution, and the optimal contact time for ammonium ion adsorption. Accordingly, the ammonium concentration was set to 75 mg NH_4_^+^/L at pH 7 (neutral) in 200 mL of distilled water at room temperature. Then, 200, 300, 400, and 500 mg of biochar were added and stirred to determine the optimal dose. Control concentrations were measured at the beginning of the experiments and then at 3 h contact time. Once the optimal dose of biochar was determined, the ideal pH of the solution was investigated, keeping the other variables constant (adsorbent dose, contact time, solution temperature). The pH values analyzed were 3, 5, 7, and 9. Finally, to analyze the adsorption of NH_4_^+^ as a function of contact time, we took several samples (initial concentration 75 mg/L NH_4_^+^, pH 9, and 500 mg biochar dose) every 10 min until an equilibrium concentration was obtained.

Batch experiments were carried out to obtain the isotherms by investigating ammonium adsorption on biochar samples with different NH_4_^+^ concentrations. These xperiments were carried out at pH 9 with 500 mg biochar dose, using 200 mL of solutions with initial concentrations of 10, 30, 50, 75, and 100 mg NH_4_^+^/L. The suspensions were stirred for 2 h at 250 rpm, then 2 mL samples were taken and filtered by 0.45 µm microporous membrane filters. NH_4_^+^-ion concentrations were measured using a Merck Spectroquant Nova 60 spectrophotometer. The amount of NH_4_^+^ adsorbed per unit mass of biochar was calculated through Equation (5):(5)$$q_{e}=\left(c_{i}-c_{e}\right)*\frac{V}{m}$$
where *q_e_* is the amount of ammonium ions adsorbed by biochar (mg/g) at equilibrium; *c_i_* and *c_e_* are the concentrations of ammonium ions in the initial and equilibrium solutions (mg/L), correspondingly; *V* is the volume of the aqueous solution (L); and *m* is the mass of biochar (g).

The NH_4_^+^ adsorption isotherms were studied using batch adsorption experiments at room temperature using different initial NH_4_^+^ concentrations and a constant biochar dose (500 mg). To determine the isotherm and kinetic models that successfully describe NH_4_^+^ adsorption, we fitted mathematical models to the isotherms by a nonlinear method applying Solver add-in command in Microsoft Excel. The best-fitting kinetic and isotherm models were selected primarily based on the value of the nonlinear correlation coefficient (*R*^2^). The chi-square (ʎ^2^) statistics also facilitated the selection. A value of ʎ^2^ close to zero implies that the selected model fits the experimental data; on the other hand, a high value of ʎ^2^ indicates that the model is inappropriate. *R*^2^ and ʎ^2^ were calculated using Equations (6) and (7), respectively:(6)$$R^{2}=\frac{\sum {\left(q_{e,cal}-q_{e,mean}\right)}^{2}}{\sum {\left(q_{e,cal}-q_{e,mean}\right)}^{2}+\sum {\left(q_{e,cal}-q_{e,exp}\right)}^{2}}$$
(7)$${ʎ}^{2}=\sum \frac{{\left(q_{e,exp}-q_{e,cal}\right)}^{2}}{q_{e,cal}}\;$$
where *q_e,exp_* (mg/g) is the amount of adsorbed NH_4_^+^ at equilibrium obtained from Equation (5), *q_e,cal_* (mg/g) is the amount of adsorbed NH_4_^+^ calculated from the model using Solver, and *q_e,mean_* (mg/g) is the mean of the *q_e,exp_* values.

### 2.10. Adsorption Models

The Solver function in Excel was used to fit the non-linear kinetic and isotherm models to the experimental data, ensuring the minimization of the sum of the squared differences between the experimental and predicted data [19]. Both pseudo-first-order (PFO) and pseudo-second-order (PSO) kinetic models were used in nonlinear form using Equations (8) and (9), respectively. Moreover, besides the classical Langmuir-isotherms and Freundlich-isotherm models, the Temkin, Brunnauer–Emmett–Teller (BET), Harkins–Jura, and Aranovich–Donohue isotherm models also were applied using Equations (10), (11), (12), (13), (14), and (15), respectively, as follows:(8)$$q_{t}=q_{e}*\left(1-e^{-k_{1}t}\right)$$
(9)$$q_{t}=\frac{qe^{2}\;*\;k_{2}\;*\;t}{1+k_{2}q_{e}t}$$
where *q_e_* and *q_t_* represent the adsorption capacities (mg/g) of the adsorbent at equilibrium and at time t (min), respectively; *k*_1_ is the first-order rate constant and *k*_2_ is the second-order rate constant.
(10)$$\frac{c_{e}}{q_{e}}=\frac{1}{q_{max}}c_{e}+\frac{1}{K\cdot q_{max}}$$
where *c_e_* is the equilibrium concentration of NH_4_^+^ (mg/L), *q_e_* is the amount of adsorbed NH_4_^+^ (mg/g), and *q_max_* (mg/g) and *K* (L/mg) are empirical constants.
(11)$$lnq_{e}=lnK_{f}+\frac{1}{n}lnc_{e}$$
where *K_f_* is the Freundlich characteristic constant [(mg/g) (L/g)^1/n^], while 1/n is the heterogeneity factor.
(12)$$q_{e}=\frac{RT}{b}\ln\left(A*c_{e}\right)$$
The parameters in the equation are defined as follows: b is the Temkin isotherm constant related to adsorbent–adsorbate interactions (L/g), A is the Temkin isotherm equilibrium binding constant, R is the universal gas constant with a value of 8.314 J/(mol·K), and T = 298 K is the reaction temperature.
(13)$$q_{e}=\frac{Q_{m}K_{S}c_{e}}{\left(1-K_{L}c_{e}\right)\left[1+\left(K_{S}-K_{L}\right)c_{e}\right]}\;$$
where *q_e_* is the total amount of adsorbed NH_4_^+^ on the biochar at equilibrium (mg/g), *Q_m_* is the amount of adsorbed NH_4_^+^ on the surface of the biochar (at the available sites, that is, monolayer) (mg/g), *c_e_* is the concentration of NH_4_^+^ at equilibrium (mg/L), *K_S_* is the equilibrium constant of monolayer adsorption (L/mg), and *K_L_* is the multilayer adsorption equilibrium constant (L/mg).
(14)$$\frac{1}{q_{e}^{2}}=\frac{B}{A}-\left(\frac{1}{A}\right)\log c_{e}$$
where *A* and *B* are constants in the Harkins–Jura model characterized by multilayer adsorption at a relatively large distance from the surface [20].
(15)$$q_{e}=f\left(c\right)*\frac{1}{{\left(1-b_{2}c_{e}\right)}^{n_{2}}}$$
The Aranovich–Donohue (AD) equation is an empirical method to fit multilayer adsorption isotherms. The equation is formed by two terms. The first describes the performance of adsorption at the first molecular layer, and the second describes multilayer adsorption [21]. The first term is the function *f(c)* that can be expressed using any model simulating a Type I isotherm. In the present work, the Langmuir model was considered as a function *f(c)* (Equation (10)), and *b*_2_ and *n*_2_ are the fitting constants for ammonium ions in the AD model. 

## 3. Results

### 3.1. Cellulose, Lignin, and Ash Contents of Banana Leaves

The cellulose, lignin, and ash contents of the precursor were 41.93 ± 0.27%, 19.43 ± 1.81%, and 2.50 ± 0.79%, respectively (Figure 1).

### 3.2. Yield of Biochar

The yield of our banana leaf-derived biochar samples was analyzed at three different pyrolysis temperatures. The results obtained show that as the temperature increases, the biochar yield decreases considerably. At 300, 400, and 500 °C, yields of 54.17%, 32.57%, and 23.96% were obtained, respectively (Figure 2). A similar trend was obtained in previous research with different raw materials [22].

### 3.3. pH of Biochar in an Aqueous Environment

The investigated biochar samples are all alkaline. With increasing pyrolysis temperature, the pH value also increases: pH values of 7.97 ± 0.54, 8.97 ± 0.40, and 9.53 ± 0.51 were obtained for the biochar samples prepared at 300, 400, and 500 °C (Figure 3).

Biochar suspensions typically have an alkaline pH, which usually increases with increasing pyrolysis temperatures due to the volatilization of organic acids and the breakdown of acidic functional groups [23].

### 3.4. Surface Charge Measurements

The surface of the non-modified biochar samples is negatively charged in the investigated pH ranges. The values decrease from −33.0 to −47.9 mV, from −33.1 to −39.2 mV, and from −32.7 to −40.8 mV for the biochar samples BC_300, BC_400, and BC_500, respectively, as the pH increases from 3 to 10. The behavior of the electrical charges on the biochar pyrolyzed at 300 °C shows a marked decrease after pH 7, reaching lower negative values than those obtained for the biochars produced at 400 and 500 °C. However, from pH 4 to 6, the obtained negative electrical charge values are higher than those of the other biochars. Figure 4, Figure 5 and Figure 6 show the zeta potentials as a function of pH for the biochars pyrolyzed at 300, 400, and 500 °C, respectively.

### 3.5. SEM Analysis

Morphology is a substantial aspect in adsorbent–adsorbate interactions. SEM images were taken of the non-modified biochars before and after the adsorption of ammonium ions. Figure 7, Figure 8 and Figure 9 show that all the materials have a rough, irregular surface.

The SEM results of the biochar samples show no discernible surface-related changes when comparing the micrographs taken before (Figure 7a, Figure 8a, and Figure 9a) and after (Figure 7b, Figure 8b, and Figure 9b) the adsorption of NH_4_^+^.

### 3.6. EDX Analysis

Elemental compositions of all biochars before and after the adsorption of ammonium ions were analyzed. The EDX results are shown in Table 1 and Figure 10, Figure 11 and Figure 12. 

The C content increases after adsorption in all cases, while N, Cl, K, and S contents decrease. In particular, O content increases for BC_400 following adsorption, but it decreases for BC_300 and BC_500.

P stays the same for BC_300, falls for BC_400, and grows for BC_500 after the adsorption process. Mg increases its presence after adsorption for biochar pyrolyzed at 300 °C, but it decreases in the other two research situations. For biochars pyrolyzed at 300 °C and 400 °C, the Si increases, but for the case of biochar at 500 °C, it drops. Ca finally rises with biochars at 300 °C and 500 °C but falls in the case of biochar at 400 °C. Based on these findings, pyrolysis temperature influences the elemental composition, affecting the properties and adsorption effectiveness of biochar. 

The decrease in the concentration of elements such as Cl, K, Ca, and S leads us to conclude that ammonium ions occupy the active sites present in the biochar while the aforementioned elements are released into the solution.

The data presented in Table 1 were used to calculate the O/C and O+N/C ratios. The results are as follows: For the BC_300, the O/C ratio is 1.496 and the O+N/C ratio is 1.597. For the BC_400, the O/C ratio is 1.193 and the O+N/C ratio is 1.266. Lastly, for the BC_500, the O/C ratio is 1.814 and the O+N/C ratio is 2.20.

### 3.7. FT-IR Analysis

Fourier transform infrared spectroscopy (FT-IR) was applied to investigate the functional groups on the surface of each biochar, and it is essential to understand the mechanisms involved in pyrolysis [3]. BC_500 lacks aromatic C–O and C–H functional groups. Table 2 and Figure 13 summarize the functional groups identified in the biochars.

### 3.8. SSA Measurements

Specific surface areas were measured based on the BET adsorption–desorption method, and the results are shown in Table 3. Figure 14, Figure 15 and Figure 16 show the pore diameter distribution and the adsorption–desorption curves of our biochars.

### 3.9. Adsorption Results

#### 3.9.1. Effect of Biochar Dose, pH, Contact Time, and Initial Concentration in Ammonium Ions Adsorption

Once the main characteristics of the biochars were analyzed, they were tested in ammonium ion adsorption experiments to determine the optimal biochar dose for NH_4_^+^ removal. For this purpose, the pH and temperature values were kept constant (pH 7 and room temperature), while the initial concentration of NH_4_^+^ was 75 mg/L and the contact time for adsorption was 80 min. The results show that the NH_4_^+^ removal percentages increase with increasing biochar dose. For the biochars prepared at 300, 400, and 500 °C, 7.86%, 12.5%, and 10.96% NH_4_^+^ removals were observed, respectively (Figure 17). 

Another series of experiments were performed to determine the optimal pH for NH_4_^+^ removal. In these experiments, the biochar dose and temperature values were kept constant (500 mg and room temperature), while the initial NH_4_^+^ concentration was 75 mg/L and the contact time for adsorption was 80 min. The results show that the amount of adsorbed NH_4_^+^ increases with increasing pH value, the maximum percentages of removed ammonium ions were 8.30%, 14%, and 16.57% with biochars pyrolyzed at 300, 400, and 500 °C, respectively. Figure 18 shows the NH_4_^+^ removal percentages at different pHs for each biochar studied. 

In the preliminary experiments, 2 mL of samples were taken every 10 min and the NH_4_^+^ concentrations were recorded for the first 60 min then samples were taken every 20 min for 2 h until a final equilibrium concentration was reached. For all cases, the optimal contact time was 60 min. The effects of contact time on the adsorption performance of the three biochars are shown in the kinetics section. 

Finally, the effect of the initial concentration of ammonium ions was analyzed and the results are shown in Figure 19.

There is a similar NH_4_^+^ removal trend for the three types of biochar, which means that the pyrolysis temperature does not affect the percentage of adsorbed NH_4_^+^.

#### 3.9.2. Adsorption Kinetics

Adsorption processes and potential rate control phases, such as mass transports and chemical reactions, can be explained using a kinetic model [24]. Pseudo-first-order (PFO) and pseudo-second-order (PSO) kinetic models were applied. Table 4 indicates the parameters of the kinetic models. 

The PFO model fits the results obtained for the BC_300 better, but for the BC_400 and BC_500, the PSO kinetic model fits the experimental data the best. Figure 20, Figure 21 and Figure 22 show the fitting of kinetic models to the experimental data.

#### 3.9.3. Adsorption Isotherms

Empirical models were used to describe the equilibrium adsorption of ammonium ions. The Langmuir, Freundlich, and Temkin models are the most common isotherm models for monolayer adsorption; however, the multilayer isotherm model can be better fitted to the results, so the BET, Harkins–Jura, and Aranovich–Donohue models were used instead. Table 5 describes the models and their respective parameters.

Only considering monolayer adsorption, the Temkin isotherm model fits the experimental data better using the biochars pyrolyzed at 300 °C and 400 °C; however, for the biochar pyrolyzed at 500 °C, the Freundlich model fits the best. On the other hand, considering the entire adsorption process, the Harkins–Jura model fits the experimental data the best for all biochars (studied for multilayer adsorption). Figure 23, Figure 24 and Figure 25 show the isotherm models fitted to the experimental data.

#### 3.9.4. Mechanisms of Ammonium Ion Removal by Biochars

To further understand the removal of NH_4_^+^ by biochars, some factors governing the adsorption process, such as specific surface area, ion exchange, surface functional group interaction, and precipitation, are illustrated in Figure 26.

## 4. Discussion

Raw materials play an important role in the characteristics of the biochar produced. The banana leaves we used contain 41.93% cellulose, 19.43% lignin, and 2.50% ash. Our results can be compared with the research of Ribas et al., who applied procedures based on the Van Soest and Wine method (ASTM E1755-01) to determine cellulose, lignin, and ash contents. After burning to a constant weight in a muffle furnace at 575 °C, their semi-dried banana leaves contained 26.7% cellulose, 17% lignin, and 8.7% ash [6].

Our banana leaf-derived biochars prepared at different pyrolysis temperatures show substantial differences, which will be analyzed in this section from the perspective of previous research regarding the influence of biochar production conditions.

Several studies have shown a decrease in the yield of biochar as the pyrolysis temperature increases. This phenomenon is attributed to various factors including the thermal cracking of heavy hydrocarbons, leading to increased liquid and gaseous products [12]. Furthermore, the decrease in biochar yield with increasing pyrolysis temperature is further explained by the loss of organic carbon resulting from the thermal decomposition of lignocellulosic biomass and the dehydration of organic components [25]. 

In our case, pyrolysis temperature and biochar yields correlate inversely: yields of 54.17%, 32.57%, and 23.96% were obtained at 300, 400, and 500 °C temperatures. Similar results were presented in the work of Ho Kim et al., where pitch pine was pyrolyzed at 300, 400, and 500 °C and the yields achieved were 60.7%, 33.5%, and 14.4%, respectively [26].

The release of minerals from the organic matrix of biochar is related to an increase in pH with increasing pyrolysis temperature because biological acids are continuously broken down during pyrolysis. Separating organic (carbon) and inorganic (alkali metal salts, ash) compounds has been found to raise the pH [14]. The biochars analyzed in this study achieved pH values of 7.97 ± 0.54 at 300 °C, 8.97 ± 0.40 at 400 °C, and 9.53 ± 0.51 at 500 °C in an aqueous environment. Similar results were reported by Xu et al., who used rice straw and pyrolysis temperatures of 300 °C, 500 °C, and 700 °C with a residence time of 2 h under a N_2_ atmosphere. The pH values achieved were 6.61 ± 0.05, 9.28 ± 0.07, and 10.06 ± 0.1, respectively [27], at 400 °C, under the same conditions, the pH was 9.58 ± 0.03 [28]. The breakdown of phenolic (–OH) groups on the surface of biochar is generally facilitated by an increase in pH, which increases the negative surface charge and thus the electrostatic attraction [29]. In addition, as the pyrolysis temperature increases, the concentration of minerals, such as magnesium (Mg), potassium (K), and calcium (Ca) tends to increase in the biochar. The presence of these minerals can contribute to the alkalinity of biochar. Biochar produced at higher temperatures (such as at 500 °C in our case) may have a higher concentration of these minerals, thus increasing its alkalinity compared with biochar produced at lower temperatures [30].

The zeta potentials are negative for all of our biochars in all pH ranges (from −33.0 to −47.9 mV for BC pyrolyzed at 300 °C, from −33.1 to −39.2 mV for BC pyrolyzed at 400 °C, and from −32.7 to −40.8 mV for BC pyrolyzed at 500 °C at pH 3 and pH 9, respectively), indicating the presence of negative surface charges. The decreasing trend is more noticeable when the pH increases, meaning that the number of negative charges increases on the biochar surface. Functional groups contribute significantly to the biochar surface charge including carboxyl and phenolic groups. The number of H^+^ decreases with the increase in pH which therefore can accelerate the deprotonation of surface groups [31]. The capacity of biochar functional groups to adsorb positive ions such as NH_4_^+^ is enhanced by the negative surface [32]. Biochar from banana leaves has been poorly researched; this is why the present work aims to investigate this type of biomass. The only comparable work is presented by Liu et al., who characterized biochar derived from banana pseudostem pyrolyzed at 200 °C for 1 h in a N_2_ atmosphere. The zeta potentials were 7 mV at pH 3 and −28 mV at pH 9, implying that the biochar had more negative charges on the surface [33].

The holes and pores formed in the structure of our biochar are due to the evolution of volatile matter during pyrolysis [34]. SEM images have shown that our biochars have a rough, irregular, and highly porous surface before and after NH_4_^+^ adsorption. The representative SEM micrographs show that the biochar samples (Figure 6a, Figure 7a, and Figure 8a) underwent aggregation due to NH_4_^+^ adsorption. The negative charges, which formed due to deprotonation, attracted the positive charge of NH_4_^+^ in aqueous solution. Similar images were presented in the study of Ramesh et al., where biochar derived from banana leaf sheath produced at 500 °C for 3 h in an N_2_ atmosphere was analyzed [35].

The composition of biochars is influenced by feedstock and pyrolysis conditions. Previous research has emphasized the importance of the elemental composition of biochar. Elements such as carbon, oxygen [36], calcium, potassium, and chlorine [37] may influence the adsorption capacity, stability, and potential environmental applications of biochar. Our biochars contain nitrogen, which is expected due to the raw material from which the biochars are derived: the leaves have relatively high protein and nitrogen contents [38]. All biochars analyzed in the present work have phosphorus originating from the raw material because it does not volatilize below 800 °C [39].

The C and O peaks indicate that the biochar samples have the possibility to contain –OH and –COOH groups in the composition of biochars [40]. The degree of aromatization of biochar increases as the pyrolysis temperature increases, whereas the amounts of hydrogen, nitrogen, and other elements (K, S) decrease. The cation exchange capacity and number of surface oxygen-containing functional groups also decrease. When the pyrolysis temperature increases, the alkalinity of the biochar also increases [41]. The results obtained show that the amount of Mg, Cl, K, and Ca increases with increasing pyrolysis temperature.

Li et al. reported that the presence of Mg in biochars can be linked to the ammonium adsorption capacity [42]. Silicon in biochars can also contribute to their adsorption properties because this element can participate in Coulombic and π–π electron–donor–acceptor interactions, which are critical for adsorption [43]. Studies by Asada et al. and Bian et al. demonstrate that the presence of K in biochar enhances its adsorption properties in aqueous solutions due to cation exchange reactions [44,45], which agrees with our results as the K amount decreases after the adsorption of ammonium ions.

The results of elemental analysis show the lack of an increase in nitrogen content in post-ammonium ions adsorption using biochar as an adsorbent can be attributed to several factors. One explanation might be that EDX can provide elemental distribution of the surface of the biochar but not the components of the pores’ surfaces [46]. A possible explanation could be that during the pyrolysis, the functional groups first diminish from the surface, and then from the pores; thus, there are more adsorption active sites in the pores. These are not easily identifiable through techniques like Energy-Dispersive X-ray Spectroscopy (EDX) or Fourier transform infrared spectroscopy (FT-IR), these methods may not specifically pinpoint the active sites responsible for interactions with ammonium ions. This explanation is proved by control measurements which show that after adsorption the dried ammonium-containing biochars release ammonia into the aqueous medium [47].

Earlier studies have shown that the functional groups of biochar are highly relevant to the physical and chemical properties, which affect the adsorption performance of the biochar [45]. Research conducted by Wang et al. and Afjeh et al. emphasizes the importance of functional groups (specifically those that include oxygen) during the adsorption of NH_4_^+^ onto the biochar surfaces [48,49]. In these terms, the potential presence of carboxyl groups may increase the ammonium ions adsorption capacity; the C–O stretching represented by peaks around 1319 cm^−1^ (BC_300) and 1317 cm^−1^ (BC_400) support the presence of carboxyl groups. Based on previous research, the presence of the C–O functional group in the biochar structure can enhance the capabilities for ammonium ions adsorption [50]. The BC_500 sample lacks this functional group, which leads to correlating the results obtained in the ammonium ion adsorption experiments in the case of BC_500, the lowest values of adsorption capacity presented; it is in accordance with the fact that the biochar, pyrolyzed at a higher temperature, has a less acidic (more alkaline) nature and it also explains the decreased ammonium adsorption capacity. 

The peaks around 791 cm^−1^ for the BC_300; 783 cm^−1^ for the BC_400; and 876 cm^−1^ for the BC_500 indicate the presence of amine and aromatic compounds N–H and C–H bending vibrations. The peaks around 1113 cm^−1^ (BC_300), 1105 cm^−1^ (BC_400), and 1115 cm^−1^ (BC_500) are related to the presence of amines, alcohols, ethers, and silicates C–N, C–O, and Si–O–Si stretching vibrations [51]. 

The peaks around 1605 cm^−1^ for the BC_300, 1614 cm^−1^ for BC_400, and 1580 cm^−1^ in BC_500 are ascribed to carboxyl groups, amines and carbonyl groups, N–H bending, C=C stretching, and C=O stretching [51].

The peaks around 2359 cm^−1^ (BC_300), 2635 cm^−1^ (BC_400), and 2363 cm^−1^ (BC_500) indicate the presence of thiols S–H stretching. Finally, the peaks around 3406 cm^−1^ (BC_300), 3424 cm^−1^ (BC_400), and 3423 cm^−1^ (BC_500) indicate hydroxyl groups (alcohols, phenols) and amines or ammonium ions O–H and N–H stretching [51].

According to our results, negatively charged organic groups of biochars can serve as adsorption sites for ammonium ions [52]. Additionally, N-containing functional groups on biochar surfaces change during the adsorption of ammonium ions, indicating chemical mechanisms involving π–π bonds and hydrogen bonding effects [53]. The stretching vibrations of the (O–H) band between 3406–3424 cm^−1^ were identified as water molecules that are physically adsorbed or trapped within the biochar matrix [54,55].

As the pyrolysis temperature increased, the band intensity progressively decreased, suggesting a prominent loss of water. The bands at 2924 cm^−1^ (aliphatic C–H stretching) disappeared at pyrolysis temperatures higher than 300 °C [43].

Prior studies have identified significant differences in how the functional groups in biochar behave at various pyrolysis temperatures. Nalaya et.al. discovered that biochar pyrolyzed at 300 °C has a greater capacity for specific molecules to be adsorbed on its surface when compared with biochar pyrolyzed at higher temperatures. This suggests that the biochar prepared at 300 °C had more polarity and oxygenated functional groups [56]. Zhao et al. investigated the effect of temperature during pyrolysis on the characteristics of biochar made from apple tree branches. The number of acidic functional groups dropped with rising pyrolysis temperature according to the study, especially for carboxylic functional groups [57]. Finally, Sizirici et al. examined the effect of temperature applied during pyrolysis on the characteristics of biochar made from date palm trash. Biochar pyrolyzed at lower temperatures has functional qualities similar to its feedstock [58].

O–H and C=C vibrations decreased with increasing pyrolysis temperatures, which may suggest that the phenolic compounds in lignin had been degraded. This can be attributed to the acceleration of dehydration reaction in biomass [14]. However, biochars produced at temperatures above 600 °C are characterized by lower amounts of H- and O- containing functional groups due to the dehydration and deoxygenation of the biomass [11].

Biochar undergoes physical modifications as the pyrolysis temperature rises, such as decreasing pore diameter and increasing specific surface area. For example, high pyrolysis temperatures decrease pore diameter and increase specific surface area [58]. For our biochars, the following results were obtained: For BC_300, the value for SSA was 3.55 m^2^/g; the total pore volume obtained was 0.0082 cm^3^/g; and the average pore diameter was 12.61 nm. For BC_400, the SSA was 6.68 m^2^/g; the total pore volume obtained was 0.014 cm^3^/g; and the average pore diameter was 9.13 nm. For BC_500, the SSA of biochar was 9.77 m^2^/g; the total pore volume obtained was 0.015 cm^3^/g; and the average pore diameter was 7.05 nm. Liu et al. carried out similar research, but they used banana leaves and stems as raw materials, and thus they obtained different results. This shows that different raw materials yield different results, and that biochar produced at a temperature of 400 °C for 3 h obtained an SSA value of 15.73 m^2^/g; the total pore volume obtained was 0.068 cm^3^/g; and the average pore diameter was 17.04 nm [59].

The macropore volume of biochar grows as the pyrolysis temperature rises because the biochar loses volatile material. This elimination of volatile substances increases the specific surface area and may decrease the average pore size [60].

A higher specific surface area per volume pore indicates a greater availability of active sites for adsorption, thereby enhancing the capacity of the biochar to capture ammonium ions from aqueous solutions [61]. The results from Table 3 of the present manuscript demonstrate that the highest value of the ratio between the specific surface area and the total pore volume corresponds to the biochar sample BC_500 (651.33m^2^/cm^3^); however, this biochar presented the lowest ammonium adsorption capacity, which leads us to conclude that, specifically for our biochar, the functionalization has a key role in the adsorption of ammonium ions.

Regarding the adsorption experiments of ammonium ions, the results obtained in the present study show that this is significantly influenced by the pH of the solution. The best performance was achieved at pH 9 with the biochar pyrolyzed at 500 °C. At lower pH values (<5), H^+^ and other cations compete with ammonium ions for adsorption sites. In contrast, at higher pH values, the NH_4_^+^/NH_3_ balance swings towards the production of electrically neutral NH_3_ [62].

The PFO kinetic adsorption model fits the experimental data best for the biochar pyrolyzed at 300 °C, suggesting that the adsorption process follows a first-order reaction concerning the concentration of ammonium ions. This indicates that the concentration of ammonium ions and the availability of adsorption sites are the primary factors influencing the rate of adsorption. This conclusion is corroborated by Li et al., who demonstrated that the PFO kinetic model offered a better fit to the experimental data for the remotion of phosphate, ammonium, and organic substances by MgO-impregnated biochar [42]. On the other hand, the PSO model fits the experimental data the best for the biochars pyrolyzed at 400 °C and 500 °C. This demonstrates that the adsorption rate is affected by the chemical interactions between the adsorbate and the adsorbent: chemical adsorption (chemisorption kinetics) occurs. This observation is supported by several previous studies. For example, Wang et al. used maple wood biochar to adsorb ammonium ions and demonstrated that the PSO kinetic model was the most suitable for describing the adsorption process, indicating a chemisorption mechanism [48]. 

The monolayer analysis results obtained at lower NH_4_^+^ concentrations (below 75 mg NH_4_^+^/L) showed that the Temkin isotherm model fits the experimental data the best, using biochar pyrolyzed at 300 and 400 °C. The results obtained for the biochar prepared at 500 °C signify that the monolayer adsorption is effective in systems with a homogeneous adsorbent surface and a uniform distribution of adsorption sites, providing valuable information on the adsorption capacity and behavior of biochar in aqueous solutions and that ammonium ions adsorb via chemisorption [63]. 

The research carried out by Zungu et al. highlighted that the Temkin isotherm model accurately describes the adsorption of salicylic acid, diclofenac, and caffeine on biochar derived from biowaste coffee grounds. This suggests that the Temkin isotherm model effectively describes the adsorption behavior of these pharmaceuticals on biochar, considering the physicochemical properties and indirect pharmaceutical interactions [64]. 

It is important to mention that the difference between the R^2^ and ʎ^2^ parameters between Temkin and Freundlich models for biochar pyrolyzed at 500 °C is negligibly small. However, the Freundlich model fits better taking into consideration the lowest chi-square value, as the R^2^ values are the same for the Temkin and Freundlich models. This suggests that NH_4_^+^ adsorption is characterized by a heterogeneous surface with a non-uniform distribution of sorption heat over the surface. Additionally, this model describes the adsorption behavior of biochar by considering multilayer adsorption and the varying affinity of the adsorbate for different sites on the biochar surface. This observation is also supported by the study conducted by Wang et al. [48]. Moreover, the work of Yin et al. on the removal of ammonium ions and phosphate using Mg-modified biochar highlighted that the Freundlich model was suitable for describing the adsorption [18].

Considering the entire NH_4_^+^ adsorption process, the Harkin–Jura isotherm model fits the experimental data the best for all biochars analyzed. This implies that the adsorption process involves multilayer adsorption on the surface of the biochars with a heterogeneous pore distribution. This model accounts for the possibility of multiple layers of adsorption and considers the non-uniform distribution of sorption sites on the biochar surface. The Harkins–Jura isotherm equation provides a foundation for comprehending the adsorption behavior in systems where adsorbate molecules can interact with various places on the adsorbent surface, resulting in the formation of numerous layers [65,66]. This conclusion is supported by the work of Galamini et al. on NH_4_^+^ adsorption in raw liquid manure using natural chabazite zeolite-rich tuff. They demonstrated that the Harkin–Jura isotherm correlated well with the experimental data, supporting the presence of multilayer adsorption. Their conclusions also reinforce the assumption of heterogeneous pore distribution and the formation of an adsorption multilayer on the biochar surface [67].

The affinity of biochar for ammonium ions can be increased by functionalization with negatively charged groups, such as carboxyl groups, through electrostatic interactions. Such functional groups have the potential to strengthen ion exchange processes [68].

## 5. Conclusions

We found that pyrolysis temperature significantly affects the physio-chemical properties of biochar derived from banana leaves and its ammonium ion adsorption behavior. Pyrolysis temperature also has an impact on the yield of biochar because of the loss of organic carbon and the increase in ash fraction with increasing pyrolysis temperature. There is a directly proportional relationship between the pH and the pyrolysis temperature: as the latter increases, the former also increases due to the separation of organic carbon and inorganic compounds. At the same time, the carbon amount, specific surface area, and pore volume also increases, but the O/C ratios decreases. However, further increasing the temperature to 500 °C increases both the O/C and (O+N)/C ratios.

The biochar pyrolyzed at 300 °C follows the pseudo-first-order kinetic adsorption model and the Temkin isotherm models considering only the monolayer phase. The biochars pyrolyzed at 400 °C and 500 °C follow the pseudo-second-order kinetic adsorption and Freundlich isotherm models. The Harkins–Jura isotherm model fits the experimental data the best for all biochars considering the entire adsorption behavior. Biochar made at 300 °C is the most efficient, achieving a capacity of 7 mg of ammonium for each gram of biochar used, while biochar prepared at 500 °C shows the lowest value of 5.6 mg/g, and the biochar pyrolyzed at 400 °C achieves 6.13 mg/g of adsorbed NH_4_^+^.

The present work describes a potential way to apply the circular economy concept using an important agricultural waste, that is, banana leaves as a feedstock to produce an environmentally friendly and low-cost adsorbent capable of removing pollutants that cause eutrophication. In the future, to further increase the adsorption capacity of biochars for the removal of ammonium ions, either the electrostatic repulsion has to be decreased or the surface contact has to be improved.

## Figures and Tables

**Figure 1 materials-17-03180-f001:**
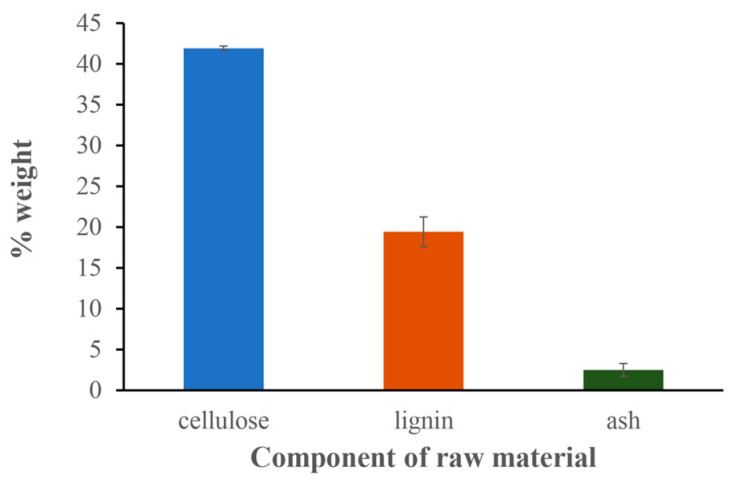
Cellulose, lignin, and ash contents in the precursor.

**Figure 2 materials-17-03180-f002:**
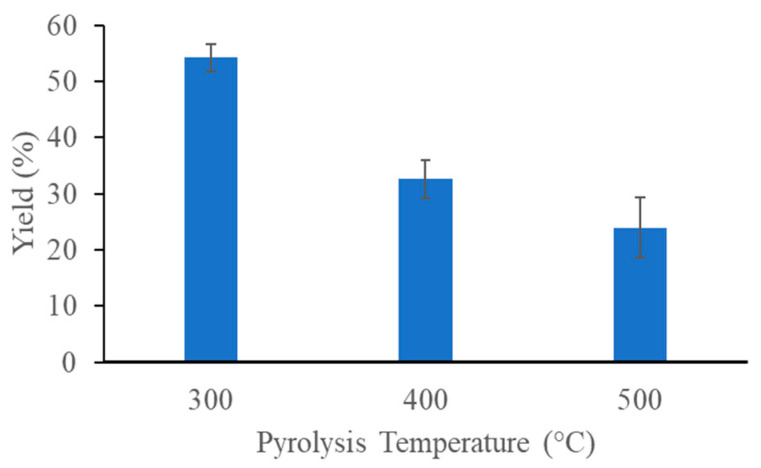
Biochar yields as a function of pyrolysis temperature.

**Figure 3 materials-17-03180-f003:**
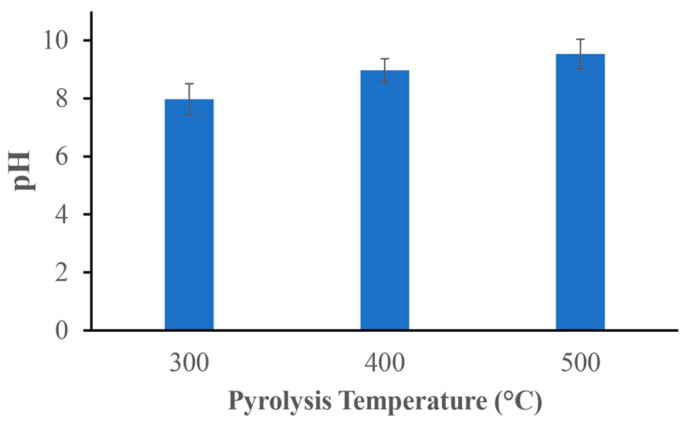
pH of the biochar suspensions produced at different pyrolysis temperatures.

**Figure 4 materials-17-03180-f004:**
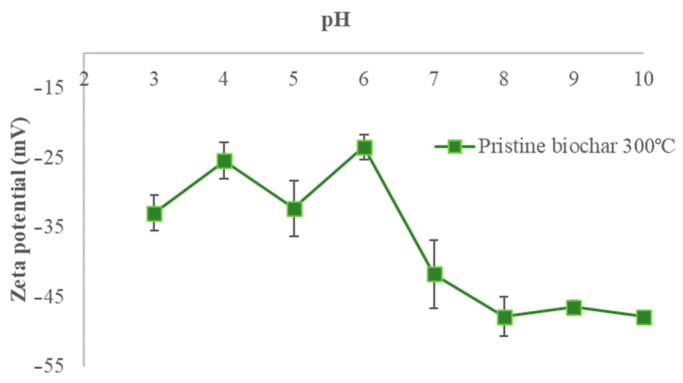
Zeta potential of BC_300 °C as a function of pH (at 0.01 M NaCl concentration).

**Figure 5 materials-17-03180-f005:**
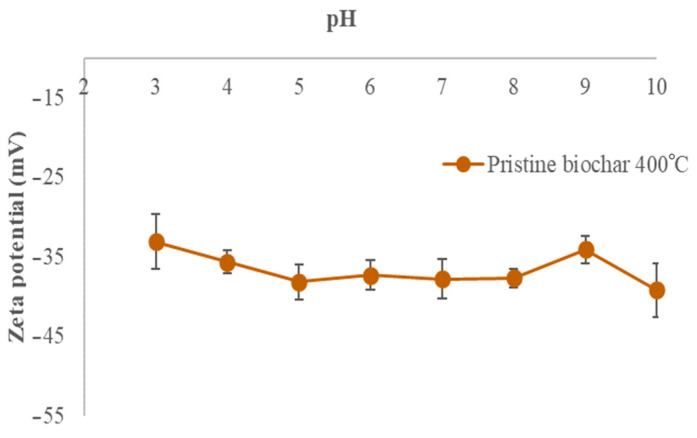
Zeta potential of BC_400 °C as a function of pH (at 0.01 M NaCl concentration).

**Figure 6 materials-17-03180-f006:**
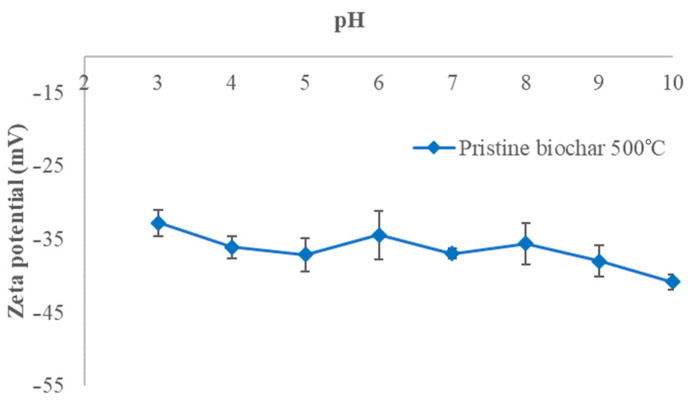
Zeta potential of BC_500 °C as a function of pH (at 0.01 M NaCl concentration).

**Figure 7 materials-17-03180-f007:**
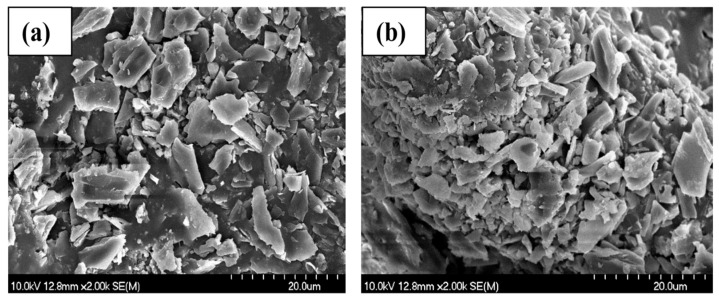
Scanning electron microscope images of BC_300 °C: (**a**) before NH_4_^+^ removal and (**b**) after NH_4_^+^ removal.

**Figure 8 materials-17-03180-f008:**
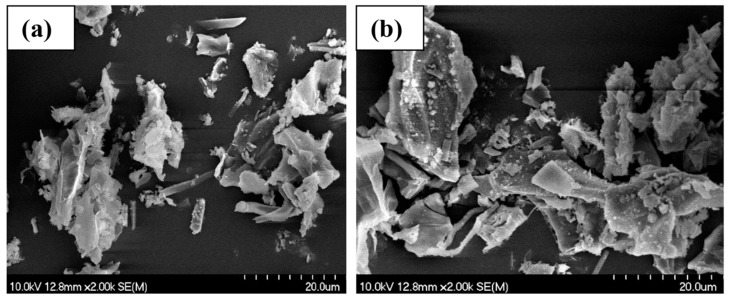
Scanning electron microscope images of BC_400 °C: (**a**) before NH_4_^+^ removal, and (**b**) after NH_4_^+^ removal.

**Figure 9 materials-17-03180-f009:**
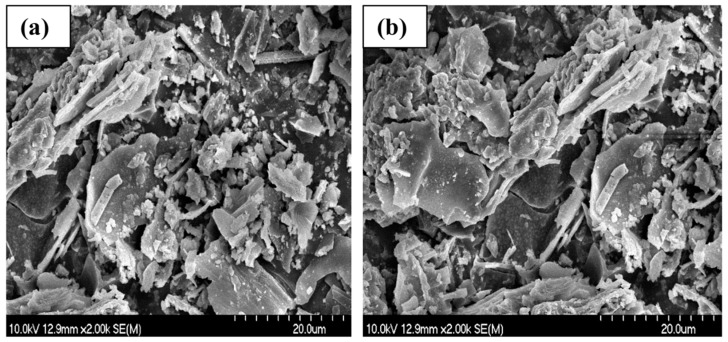
Scanning electron microscope images of BC_500 °C: (**a**) before NH_4_^+^ removal, and (**b**) after NH_4_^+^ removal.

**Figure 10 materials-17-03180-f010:**
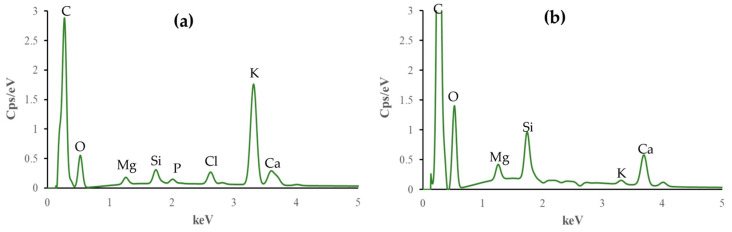
Elemental composition of non-modified biochar pyrolyzed at 300 °C (**a**) before and (**b**) after NH_4_^+^ removal.

**Figure 11 materials-17-03180-f011:**
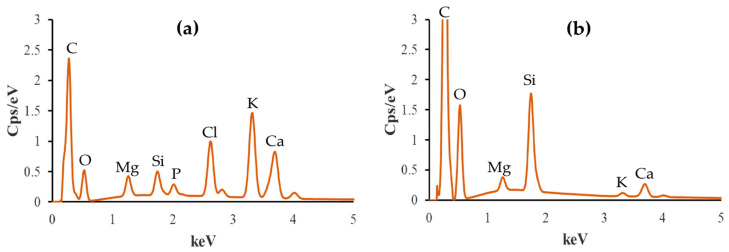
Elemental composition of non-modified biochar pyrolyzed at 400 °C (**a**) before and (**b**) after NH_4_^+^ removal.

**Figure 12 materials-17-03180-f012:**
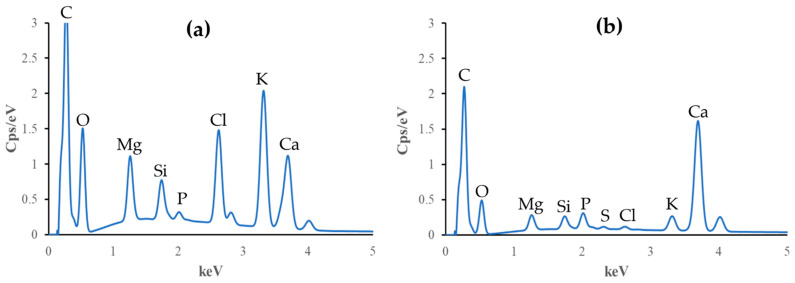
Elemental composition of non-modified biochar pyrolyzed at 500 °C (**a**) before and (**b**) after NH_4_^+^ removal.

**Figure 13 materials-17-03180-f013:**
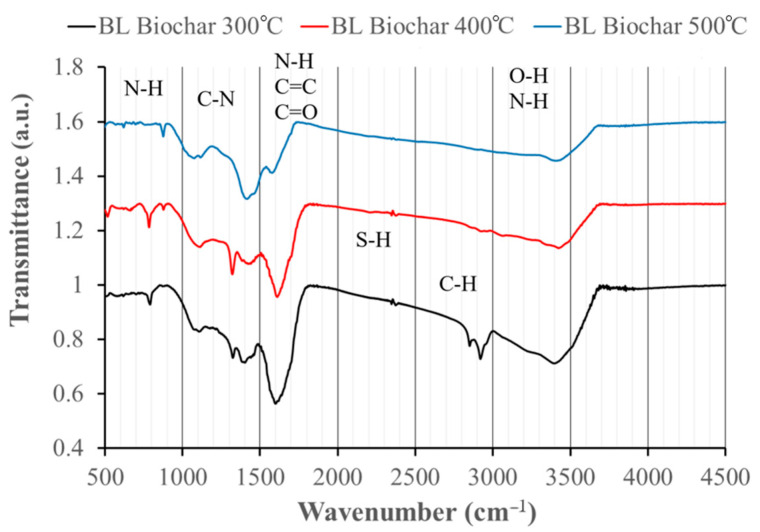
FT-IR spectra of the biochars.

**Figure 14 materials-17-03180-f014:**
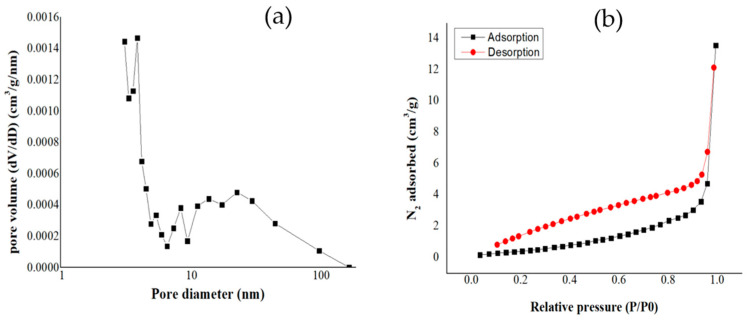
(**a**) Pore diameter and (**b**) adsorption–desorption curves for biochar pyrolized at 300 °C.

**Figure 15 materials-17-03180-f015:**
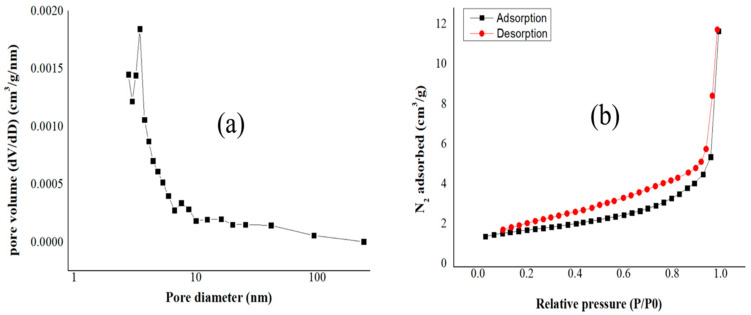
(**a**) Pore diameter and (**b**) adsorption–desorption curves for biochar pyrolized at 400 °C.

**Figure 16 materials-17-03180-f016:**
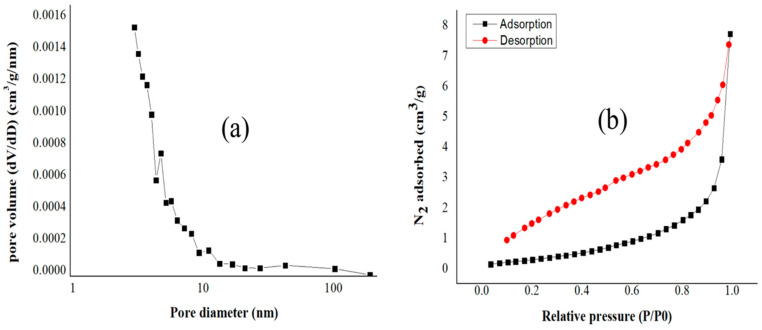
(**a**) Pore diameter and (**b**) adsorption–desorption curves for biochar pyrolized at 500 °C.

**Figure 17 materials-17-03180-f017:**
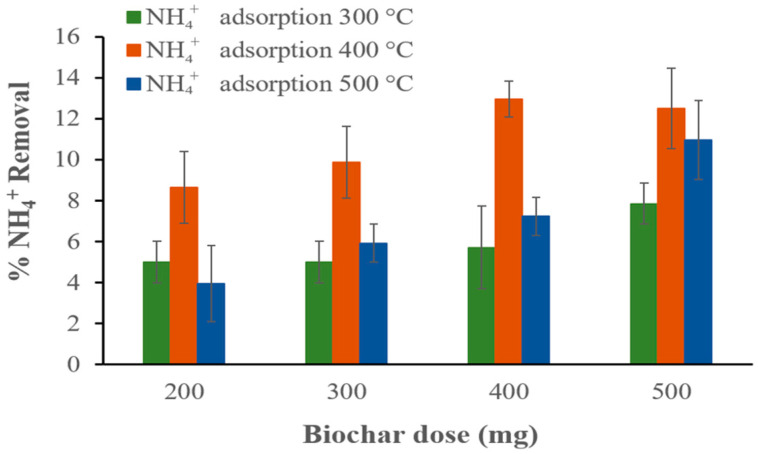
Effect of biochar dose on NH_4_^+^ removal in an aqueous solution containing an initial NH_4_^+^ concentration of 75 mg NH_4_^+^/L at pH 7 and room temperature.

**Figure 18 materials-17-03180-f018:**
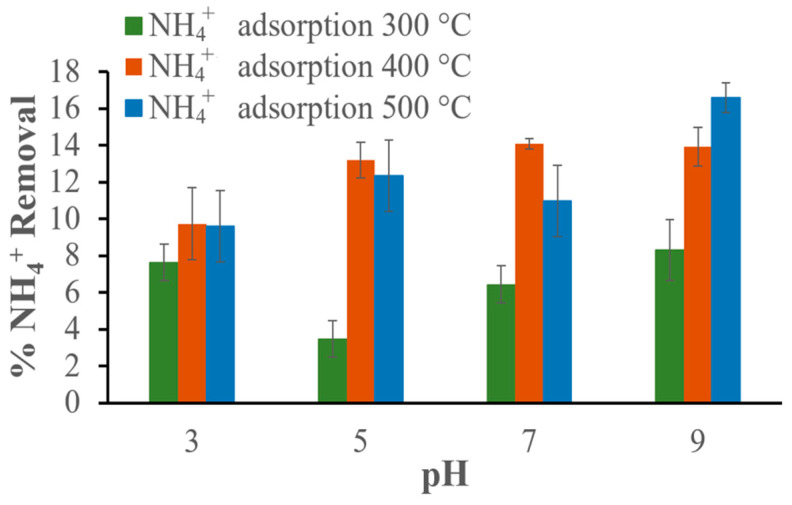
Effect of pH on NH_4_^+^ removal in an aqueous solution with an initial concentration of 75 mg NH_4_^+^/L and 500 mg of biochar.

**Figure 19 materials-17-03180-f019:**
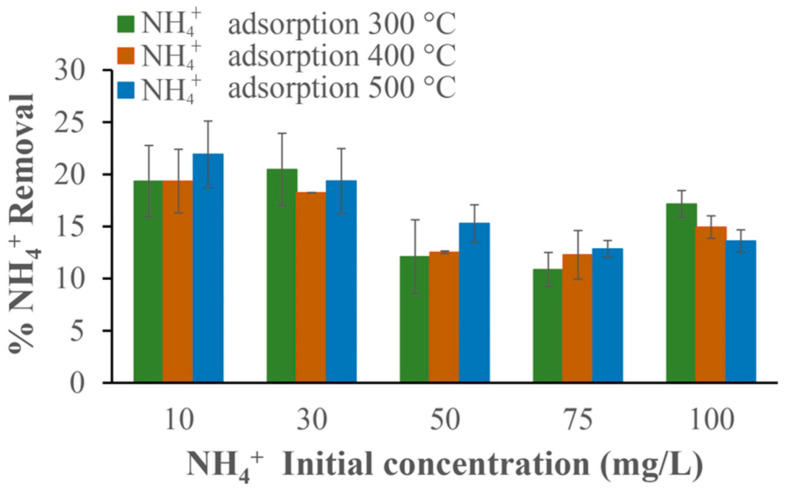
Effect of the initial concentration of NH_4_^+^ with 500 mg of biochar and at pH 9.

**Figure 20 materials-17-03180-f020:**
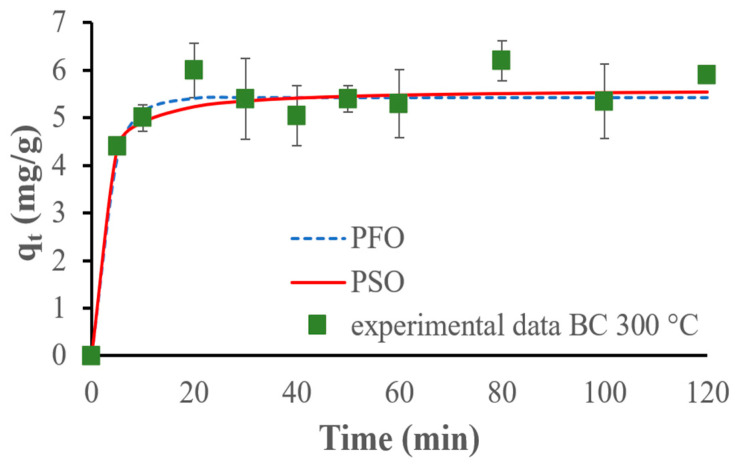
PFO and PSO kinetic models fitted for experimental NH_4_^+^ adsorption results, using 500 mg of the biochar pyrolyzed at 300 °C and 75 mg/L of initial ammonium ion concentration.

**Figure 21 materials-17-03180-f021:**
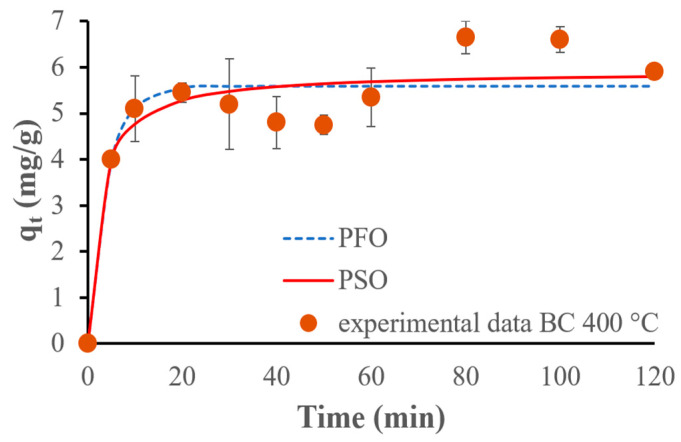
PFO and PSO kinetic models fitted for experimental NH_4_^+^ adsorption results, using 500 mg of the biochar pyrolyzed at 400 °C and 75 mg/L of initial ammonium ion concentration.

**Figure 22 materials-17-03180-f022:**
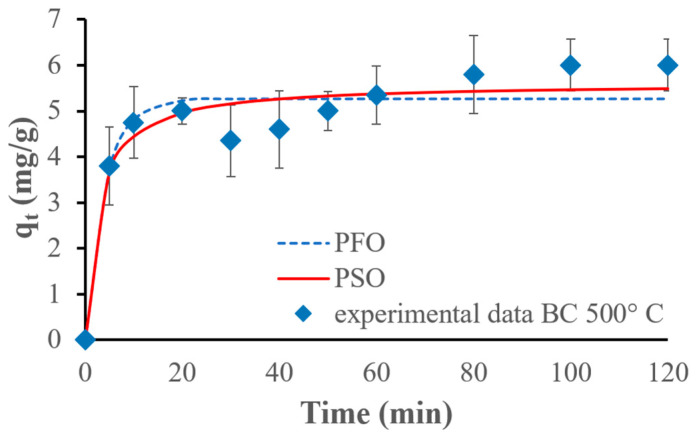
PFO and PSO kinetic models fitted for experimental NH_4_^+^ adsorption results, using 500 mg of the biochar pyrolyzed at 500 °C and 75 mg/L of initial ammonium ion concentration.

**Figure 23 materials-17-03180-f023:**
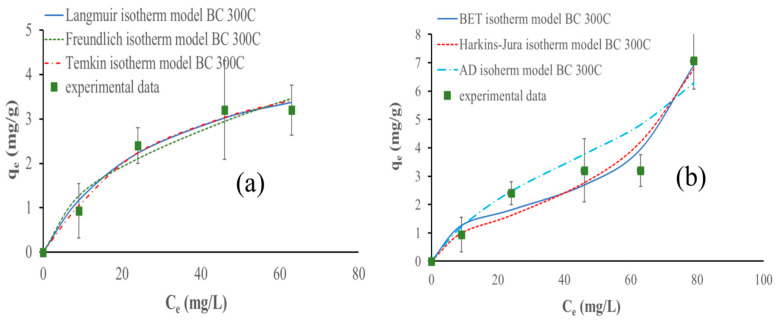
Isotherm models fitted to data obtained during NH_4_^+^ adsorption with 500 mg of biochar pyrolyzed at 300 °C at different initial concentrations of NH_4_^+^ and pH 9 in 80 min: (**a**) monolayer adsorption models and (**b**) multilayer adsorption models.

**Figure 24 materials-17-03180-f024:**
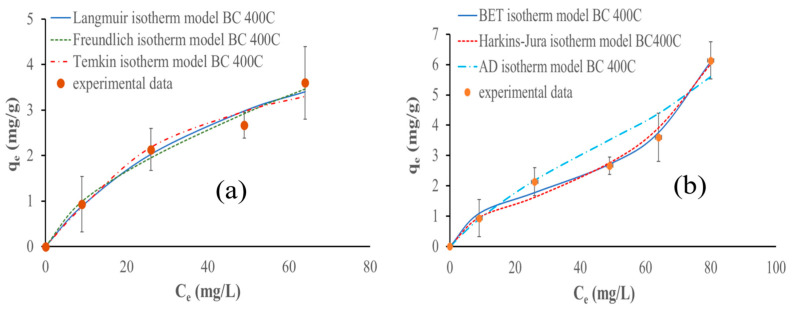
Isotherm models fitted to data obtained during NH_4_^+^ adsorption with 500 mg of biochar pyrolyzed at 400 °C at different initial concentrations of NH_4_^+^ and pH 9 in 80 min: (**a**) monolayer adsorption models and (**b**) multilayer adsorption models.

**Figure 25 materials-17-03180-f025:**
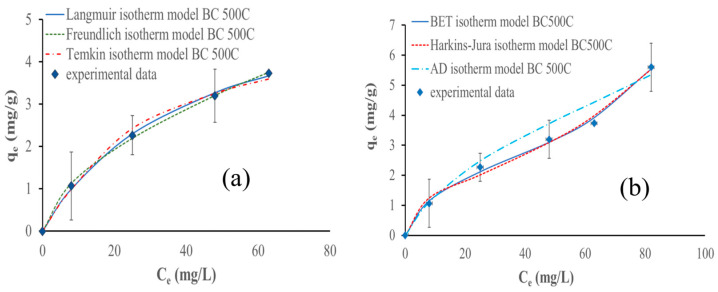
Isotherm models fitted to data obtained during NH_4_^+^ adsorption with 500 mg of biochar pyrolyzed at 500 °C at different initial NH_4_^+^ concentrations and pH 9 in 80 min: (**a**) monolayer adsorption models and (**b**) multilayer adsorption models.

**Figure 26 materials-17-03180-f026:**
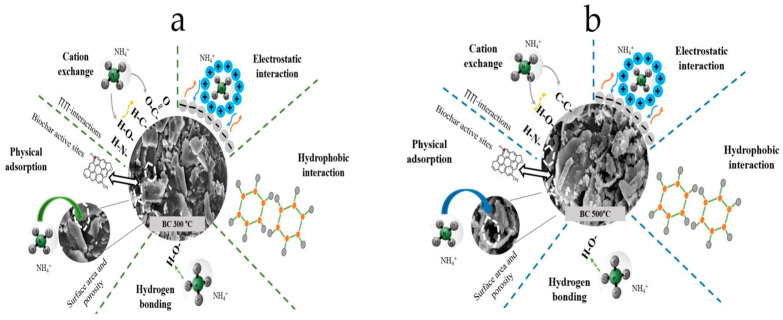
(**a**) Mechanisms suggested for the adsorption of ammonium ions using biochar pyrolyzed at 300 °C and (**b**) suggested mechanisms for adsorption of ammonium ions using biochar pyrolyzed at 500 °C.

**Table 1 materials-17-03180-t001:** Elemental composition of the biochar materials obtained at various pyrolysis temperatures.

Elemental Content (%)	Biochar					
	300 °C ba *	300 °C aa **	400 °C ba	400 °C aa	500 °C ba	500 °C aa
C	38.92 ± 4.43	45.20 ± 6.02	36.17 ± 4.31	38.70 ± 4.33	29.29 ± 2.76	39.24 ± 4.94
N	3.57 ± 0.28	3.99 ± 0.26	2.64 ± 0.15	0.0086 ± 0	11.42 ± 0.48	4.55 ± 0.39
O	58.21 ± 12.58	52.65 ± 10.83	43.15 ± 7.77	58.28 ± 12.45	53.13 ± 10.25	49.63 ± 9.93
Mg	0.21 ± 0	0.54 ± 0	1.10 ± 0	0.36 ± 0	1.49 ± 0.005	0.93 ± 0
Si	0.07 ± 0	0.72 ± 0	0.90 ± 0	2.11 ± 0	0.66 ± 0	0.51 ± 0
P	0.048 ± 0	0.04 ± 0	0.39 ± 0	0.11 ± 0	0.13 ± 0	0.59 ± 0
Cl	0.030 ± 0	0	1.80 ± 0.01	0	1.52 ± 0.01	0.09 ± 0
K	2.4 ± 0.015	0.12 ± 0	3.17 ± 0.03	0.12 ± 0	2.90 ± 0.02	0.46 ± 0
Ca	0.11 ± 0	0.74 ± 0	1.92 ± 0.016	0.43 ± 0	1.88 ± 0.013	3.92 ± 0.04
S	0.28 ± 0.0011	0	0.041 ± 0	0	0.028 ± 0	0

* Before NH_4_^+^ adsorption and ** after NH_4_^+^ adsorption.

**Table 2 materials-17-03180-t002:** Functional groups identified in the investigated biochars.

Wavenumber (cm^−1^)				
BC_300 °C	BC_400 °C	BC_500 °C	Functional Groups	Reference
791	783	876	N–H, C–H	[24]
1113	1105	1115	C–N, C–O, Si–O–Si	[24]
1319	1317	-	C–O, or C–H	[24]
1393	1418	1416	N–H, C–C	[24]
1605	1614	1580	N–H, C=C, C=O	[24]
2359	2365	2363	S–H	[24]
2924	-	-	C–H, –CH_3_ or –CH_2_	[24]
3406	3424	3423	O–H and N–H	[24]

**Table 3 materials-17-03180-t003:** BET results.

Pyrolysis Temperature (°C)	SSA (m^2^/g)	Average Pore Size (nm)	Total Pore Volume (cm^3^/g)	SSA/TPV (m^2^/cm^3^)
300	3.55	12.61	0.0082	432.93
400	6.68	9.13	0.014	477.14
500	9.77	7.05	0.015	651.33

**Table 4 materials-17-03180-t004:** Kinetic parameters for NH_4_^+^ adsorption.

Kinetic Model	Parameters	BC 300 °C	BC 400 °C	BC 500 °C
Pseudo-first order	q_e_ (mg/g)	5.431	5.593	5.267
	k_1_	0.293	0.248	0.246
	R^2^	0.775	0.603	0.498
	ʎ^2^	0.026	0.070	0.055
Pseudo-second order	q_e_ (mg/g)	5.606	5.919	5.613
	k_2_	0.125	0.070	0.068
	R^2^	0.772	0.659	0.639
	ʎ^2^	0.027	0.060	0.039

**Table 5 materials-17-03180-t005:** Isotherm models parameters for NH_4_^+^ adsorption with biochars derived from banana leaves.

Pyrolysis Temperature	Isotherm Model	Parameters	R^2^	ʎ^2^
300 °C	Langmuir	K = 0.034	0.958	0.012
		q_max_ = 4.941		
		R_L_ = 0.592		
	Freundlich	K_f_ = 0.408	0.902	0.028
		1/n = 0.517		
	Temkin	b = 2025.90	0.996	0.012
		A = 0.26		
	BET	q_m_ = 1.52	0.936	0.0795
		K_l_ = 0.0099		
		K_s_ = 0.31		
	Harkins–Jura	A = 2.192	0.975	0.201
		B = 2.059		
	Aranovich–Donohue	b_2_ = 0.01	0.825	0.195
		n_2_ = 0.355		
400 °C	Langmuir	K = 0.018	0.960	0.013
		q_max_ = 6.332		
		R_L_ = 0.734		
	Freundlich	K_f_ = 0.245	0.968	0.011
		1/n = 0.637		
	Temkin	b = 1992.90	0.993	0.020
		A = 0.22		
	BET	q_m_ = 1.56	0.988	0.0117
		K_l_ = 0.0094		
		K_s_ = 0.18		
	Harkins–Jura	A = 2.176	0.994	0.041
		B = 2.084		
	Aranovich–Donohue	b_2_ = 0.01	0.895	0.099
		n_2_ = 0.25		
500 °C	Langmuir	K = 0.025	0.996	0.001
		q_max_ = 6.038		
		R_L_ = 0.670		
	Freundlich	K_f_ = 0.337	0.998	0.001
		1/n = 0.582		
	Temkin	b = 1958.70	0.998	0.005
		A = 0.27		
	BET	q_m_ = 2.22	0.993	0.0048
		K_l_ = 0.0075		
		K_s_ = 0.11		
	Harkins–Jura	A = 3.276	0.997	0.017
		B = 2.210		
	Aranovich–Donohue	b_2_ = 0.01	0.834	0.152
		n_2_ = 0.18		

## Data Availability

Data are contained within this article.

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
