# Peer review of "Impact of Pyrolysis Temperature on the Physical and Chemical Properties of Non-Modified Biochar Produced from Banana Leaves: A Case Study on Ammonium Ion Adsorption"

_materials, 2024, doi:10.3390/ma17133180_

Round 1
Reviewer 1 Report
Comments and Suggestions for Authors
This work has studied the impact of pyrolysis temperature on the physical and chemical properties of pristine biochar produced from banana leaves, and the performance of ammonium ion adsorption was investigated. The results are valuable and can be accepted after some revisions, specific comments are as below:
1. The authors chose banana leaves as raw material, and obtained the biochar. Did the authors study other biomass and compare their properties of biochar? The work would be more significant if they are compared and analyzed.
2. The banana leaves were pyrolyzed at 300℃, 400℃ and 500℃, why not chose and added higher temperature? How the properties of biochar changed at higher temperature? The reasons for this temperature range should be given.
3. Why the authors not chose element analyzer to study the compositions of biochar, it would be more accurate to analyze the contents of element C, H, N and S.
4. The data of biochar obtained at 500℃ was missing in section 3.6, and the table would be clearer to compare the data.
5. There are still some mistakes in the format, please check and revise.
Comments on the Quality of English LanguageThere are still some mistakes in the format, please check and revise.
Reviewer 2 Report
Comments and Suggestions for Authors
Manuscript ID: materials-3014054
Title: Impact of pyrolysis temperature on the physical and chemical properties of pristine biochar produced from banana leaves: a case study on ammonium ion adsorption
Overview:
Authors produce and analyze the oxygenated biochar from banana leaves. Although banana-peel-derived biochar is relatively well-studied, the opportunity to derive activated carbon from leaves without chemical activation is relatively less investigated. The structural studies are mostly scientifically coherent, while the assessment of various absorption isotherms sheds a new light on the water cleaning possibilities with investigated biochar. However, many aspects of the paper have a considerable room for improvement, and I recommend its major revision.
Major comments:
1) Line 66 states that “Until now, no one has ever investigated the production pf biochar from banana leaves”. However, brief search indicates several papers devoted to this topic [https://www.nature.com/articles/s41598-022-05652-7; https://iopscience.iop.org/article/10.1088/1742-6596/1979/1/012003/meta]. Authors should revise the novelty statement and compare their results with previously-reported ones.
2) Currently, Harkins-Jura isotherm is considered inferior regarding the assessment of water contaminant adsorption [https://doi.org/10.1016/j.molliq.2022.121122], as it has been developed as a model of multilayer gas adsorption rather than the one suitable for aqueous media. However, in the current study it fits the analyzed data better than other isotherms. Could you comment on how that is possible: is the modification suggested in eq. 11 somehow alleviates HJ model drawbacks?
3) Section 3.5: although authors observe no S content in EDX spectra, they attribute ~2360 cm-1 FTIR feature to S-H bond. How is that possible?
4) Section 3.6, why don’t authors analyze the porosity the sample annealed at 500C? Its NH4 absorption is discussed later, so its specific area seems important for subsequent discussion.
5) Section 4 (Discussion) seems quite general. Although rigorous structural investigations are carried out, authors haven’t found out any interplay between the materials’ structure, sorption performance and annealing parameters. I encourage authors to discuss in more detail how annealing temperature influences the rearrangement of biopolymers and why 400C-annealed biochar had the best adsorption characteristics: is it mostly related to surface to volume ratio or to the functionalization with negatively-charged groups? Which chemical groups are expected to have the most effect on NH4+ removal?
Minor comments:
6) In the Title, “pristine biochar” term seems misleading, as produced biochar has several types of heteroatoms significantly affecting the materials’ properties.
7) Line 45, revise “characterice”.
8) Section 2.1: in what atmosphere were drying and pyrolysis carried out?
9) In eq. 11, what do you mean by “𝑛2”? Is it “n_2”, “n^2” or “2n”? By “q” and “c”, did you mean “q_e” and “c_e”?
10) Section 3.2: as “pH” is a characteristic of the aqueous media, I suggest to revise the term “pH of biochar”.
11) Section 3.4, Table 1: to me, it is unclear why post-adsorption samples demonstrate no increase in nitrogen content. Were the powders cleaned/heated after the experiments or NH4 desorbed by itself?
12) In Section 3.4, consider discussing the difference in the elemental composition of pre-/after-adsorption samples in more detail.
13) Line 290, “lacks the presence of aromatic C-O and C-C functional groups”: how can carbon samples lack C-C groups? Maybe you meant “C-H”?
14) The positioning of ~1600 cm-1 line at ~1620 cm-1 is rather atypical for C=C. Can the position shift be related to the formation of C=O bonds [10.3390/jcs7070264]?
15) As for the O-H bonding, do authors expect a presence of hydroxide groups on the sample surface, or its presence is related to the ambient water adsorption [10.3390/jcs7070264]?
16) Revise the Y scale title in Fig. 13(a) (dV/d?).
17) In the Y scale title in Fig. 13(b), what is STP?
18) In Conclusions, authors state that “the ratios of H/C, O/C, and (O+N)/C decreased”. However, the variations of these ratios were not discussed in the main text. Consider supplementing the manuscript. Notably, authors don’t have a direct way to assess H/C ratio, therefore I suggest to elaborate on the fragment in question.
Comments on the Quality of English LanguageI have no major concerns regarding English language. However, authors are encouraged to fix the typos.
Examples "production pf biochar" (line 66), "characterice" (line 45).
Reviewer 3 Report
Comments and Suggestions for Authors
Dear Authors,
You have shown a lot of results. The topic of the work is interesting. However, there are many similar works in the literature. I did not notice any novelty in your work. I would also like to make some remarks and suggestions.
Modify the title of the paper
What does it mean: pristine biochar?
Why did you do, for example, banana leaves and you are not a country with a banana climate? what kind of banana
Paragraph 2.1, during pyrolysis, which gas did you use in an inert atmosphere? And the amount of gas?
Why didn't you go above 500C?
Compare Figure 1 with the literature and other yields of different biomass precursors at the same or similar conditions.
The same also applies to picture 2.
Paragraph 3.3 provides a more detailed and extensive analysis. I don't see any drastic difference between a and b
In paragraph 3.4 where you presented the elements, why didn't you do an elemental analysis for CHNOS? the method is more reliable for biomass testing.
Also, do the content of lignin, hemicellulose and lignin for you precursor and see what effect it has on the final product and application
Technically fix image 12. And analyze FTIR more thoroughly
Results of paragraph 3.6. SSA measurements tabular display. You need to better describe this analysis and relate it to the functional groups and your sorptions.
Among other things, determine the type of isotherm curve.
Why didn't you do the fitting according to Temkin's model?
The discussion is better if you worked under each method and then gave the final seal in the conclusion.
You need to better connect the methods and analysis you have done.
Round 2
Reviewer 1 Report
Comments and Suggestions for Authors
The authors have responded the comments, however, the revisions are not so significant such as comment 1, comment 2 and comment 3.
More significant revisions are expected to be added.
Reviewer 2 Report
Comments and Suggestions for Authors
Manuscript ID materials-3014054
Title Impact of pyrolysis temperature on the physical and chemical properties of pristine biochar produced from banana leaves: a case study on ammonium ion adsorption
Revision 1
Authors have made a considerable improvements to the manuscript. However, there is still a room for improvement.
Question 5 is not answered in a satisfactory way.
Line 649, authors claim that "specifically for our biochar, the
functionalization with negatively-charged groups is more relevant in the adsorption of
ammonium ions." However, Zeta-potential analysis shows that samples have a comparable potential, which does't allow to derive any explicit information about the biochar charge. Also authors don't discuss the variation of FTIR spectra in a detailed enough way to draw any conclusions (for example, see question 14 from revision 0: authors implicitly confirmed that 300C and 400C samples demonstrate a considerable C=O line, while 500C doesn't, but haven't discussed it anyhow), although FTIR provides an information about polar groups "by design".
Could authors elaborate on what negatively-charged groups play a role in the ammonia sorption and how can authors detect their variation? I suggest to find some metrics which change correlates with adsorption capabilitites and discuss this correlation/dependence in a detailed way.
